# Classification of Skin Cancer Using Novel Hyperspectral Imaging Engineering via YOLOv5

**DOI:** 10.3390/jcm12031134

**Published:** 2023-02-01

**Authors:** Hung-Yi Huang, Yu-Ping Hsiao, Arvind Mukundan, Yu-Ming Tsao, Wen-Yen Chang, Hsiang-Chen Wang

**Affiliations:** 1Department of Dermatology, Ditmanson Medical Foundation Chiayi Christian Hospital, Chiayi 60002, Taiwan; 2Department of Dermatology, Chung Shan Medical University Hospital, No. 110, Sec. 1, Jianguo N. Rd., South District, Taichung City 40201, Taiwan; 3Institute of Medicine, School of Medicine, Chung Shan Medical University, No. 110, Sec. 1, Jianguo N. Rd., South District, Taichung City 40201, Taiwan; 4Department of Mechanical Engineering, Advanced Institute of Manufacturing with High Tech Innovations (AIM-HI) and Center for Innovative Research on Aging Society (CIRAS), National Chung Cheng University, No. 168, University Rd., Min Hsiung, Chiayi 62102, Taiwan; 5Department of General Surgery, Kaohsiung Armed Forces General Hospital, No. 2, Zhongzheng 1st Rd., Lingya District, Kaohsiung 80284, Taiwan; 6Hitspectra Intelligent Technology Co., Ltd., 4F, No. 2, Fuxing 4th Rd., Qianzhen District, Kaohsiung 80661, Taiwan; 7Department of Medical Research, Dalin Tzu Chi General Hospital, No. 2, Min-Sheng Rd., Dalin Town, Chiayi 62247, Taiwan

**Keywords:** skin cancer, hyperspectral imaging, convolutional neural network, YOLO

## Abstract

Many studies have recently used several deep learning methods for detecting skin cancer. However, hyperspectral imaging (HSI) is a noninvasive optics system that can obtain wavelength information on the location of skin cancer lesions and requires further investigation. Hyperspectral technology can capture hundreds of narrow bands of the electromagnetic spectrum both within and outside the visible wavelength range as well as bands that enhance the distinction of image features. The dataset from the ISIC library was used in this study to detect and classify skin cancer on the basis of basal cell carcinoma (BCC), squamous cell carcinoma (SCC), and seborrheic keratosis (SK). The dataset was divided into training and test sets, and you only look once (YOLO) version 5 was applied to train the model. The model performance was judged according to the generated confusion matrix and five indicating parameters, including precision, recall, specificity, accuracy, and the F1-score of the trained model. Two models, namely, hyperspectral narrowband image (HSI-NBI) and RGB classification, were built and then compared in this study to understand the performance of HSI with the RGB model. Experimental results showed that the HSI model can learn the SCC feature better than the original RGB image because the feature is more prominent or the model is not captured in other categories. The recall rate of the RGB and HSI models were 0.722 to 0.794, respectively, thereby indicating an overall increase of 7.5% when using the HSI model.

## 1. Introduction

The most common and most diagnosed among the cancers is nonmelanoma skin cancer (NMSC) [1]. Common types of skin cancer include basal cell carcinoma (BCC), squamous cell carcinoma (SCC), and seborrheic keratosis (SK) [2]. BCC is the least aggressive NMSC and can be characterized by cells that resemble epidermal basal cells, while squamous cells are invasive and may metastatically proliferate in an abnormal manner [3]. BCC is the most prevalent kind of skin cancer and usually develops in the head and neck area in 80% of patients without metastasizing [4,5]. Meanwhile, SCC is the second most common skin cancer, and it is also associated with aggressive tumors with highly invasive properties [6].

The demand for rapid prognosis of skin cancer has grown because of the steadily increasing incidence of skin cancer [7,8,9,10,11,12]. Early diagnosis is a critical factor in skin treatment [13,14,15]. Although doctors typically adopt biopsy, this method is slow, painful, and time consuming [16] because it collects samples from potential spots of skin cancer, and these samples are medically determined to detect cancer cells [17]. However, many computer-aided detection (CAD) models have been developed in recent years since the development of artificial intelligence (AI) to detect various types of cancers [18,19,20,21].

Haenssle et al. [22] used the InceptionV4 model to detect skin cancer and then compared the results with the findings of 58 dermatologists. The CAD model outperformed the dermatologists in sensitivity by 8% but underperformed in specificity by 9%. Han et al. [23] used ResNet-152 to classify images in an Asan training dataset, which was only at par with the results of 16 dermatologists. Fujisawa et al. [24] utilized a deep learning method on 6000 images to detect malignant and benign cases and achieved an accuracy of 76%. In addition to CAD, biosensors are a cost-effective tool for early cancer diagnosis [23,24,25]. However, environmental adaptabilities of these nanomaterials remain a challenge [26,27]. CAD modes based on conventional red, green, and blue (RGB) have also reached an upper saturation limit [28]. However, hyperspectral imaging (HSI) is an approach that can overcome the disadvantages of conventional methods [29,30].

HSI is an emerging method that analyzes a wide range of wavelengths instead of only providing each pixel one of the three primary colors [31,32]. HSI has been used in numerous classifications fields, such as agriculture [33], cancer detection [34,35,36,37], military [38], air pollution detection [29,39], remote sensing [40], dental imaging [41], environment monitoring [42], satellite photography [43], counterfeit verification [44,45,46], forestry monitoring [47], food security [48], natural resource surveying [49], vegetation observation [50], and geological mapping [51].

Many researches have also been conducted on skin cancer detection using HSI [52,53,54]. Leon et al. proposed a method based on supervised and unsupervised learning techniques for automatic identification and classification of the pigmented skin lesions (PSL) [55]. Courtenay et al. built an ad-hoc platform combined with a visible-near infra-red (VNIR) hyperspectral imaging sensor to find an ideal spectral wavelength to distinguish between normal skin and cancerous skin [56]. In another study by Courtenay et al. HSI was utilized to obtain spectral differences between benign and cancerous dermal tissue specimens [57]. However, most of current methods use an HSI imaging sensor or camera to capture the skin which is expensive and a difficult procedure.

Therefore, a rapid skin cancer detection method using the you only look once (YOLO) version 5 model combined with HSI technology was proposed in this study. The developed model was judged on the basis of five indicators, namely, precision, recall, specificity, accuracy, and F1-score.

## 2. Methods

### 2.1. Data Preprocessing

This study focuses on three skin-related diseases: BCC, SCC, and SK. The dataset consists of a training set of 654, 336, and 480 images, and a validation set of 168, 90, and 126 images of BCC, SCC, and SK, respectively. A total of 1470 images in the training set and 384 images in the validation set are trained on the network. The whole study can be divided into three parts, namely, image preprocessing, database creation, and evaluation of the training results of the YOLOv5 model, as shown in Figure 1. ISIC data were used. Figure 2, Figure 3 and Figure 4 show the squamous cell carcinoma (SCC), basal cell carcinoma (BCC), and seborrheic keratosis, respectively.

Each image was set to a default size of 640 × 640 pixels during image preprocessing to avoid problems, such as insufficient computer memory, during spectrum conversion and to ensure that the format is consistent. LabelImg software was adopted to mark the three sets of images, which are outputted as an xml file. This file is then converted to a txt file and fed to the YOLOv5 model for training. Note that the annotated image file was converted into an HSI narrow-band image (NBI) through a spectrum conversion algorithm prior to training. The three types of skin cancer conditions are first analyzed by comparing the normalized reflection spectrum of the lesion area, as shown in Figure 5. Wavelength bands with large differences in reflected light intensity are compared. The spectrum is converted into an HSI with a spectral resolution of 1 nm in the range of 380–780 nm. However, dimensional data will result in high computing and storage space requirements. Principal component analysis (PCA) is utilized to counter this problem and transform the data from the image to a new data space in a linear manner. PCA generates new component information, extracts important parts of features, and reduces the dimensionality of the image. Data are finally projected in a low-dimensional space in which spectral information is decorrelated.

Characteristic bands of the visible light absorption spectrum for skin irradiation are between 405–435 nm and 525–555 nm due to the absorption intensity and wavelength position of visible light by heme in the blood flow change. Wavelengths in the range of 435 and 525–555 nm are used to extract important parts of skin lesions, and the output obtains the HSI-NBI image, as shown in Figure 6.

Two data sets were obtained using the RGB and HSI images. The training and test sets need to be divided. Hence, 80% of the data was divided for training while the remaining 20% of the data were used for testing the model after image amplification of the two data sets. The PyTorch deep learning framework was built using the Windows 10 operating system, and the program written in the Python language was based on the Python 3.9.12 platform.

### 2.2. YOLOv5 Model

YOLOv5 was specifically chosen in this study because previous research suggests that when compared with other models such as RetinaNet or SSD, YOLO had a better detection speed which helps to achieve real-time performance [58]. YOLOv5 comprises three main parts, namely, backbone, neck, and head terminals (detailed information on its structure is presented in Appendix A). Backbone is the architecture of the convolutional neural network (CNN) and mainly composed of models, such as focus, CONV-BN-Leaky ReLU (CBL), cross stage partial (CSP), and spatial pyramid pooling (SPP) models. The function of focus is to slice the input image, reduce the required CUDA memory and number of layers, increase forward and back propagation speed, aggregate different fine-grained images, and form image features. Focus cuts the input image with a preset image of 640 × 640 × 3 into four 320 × 320 × 3 images and then passes through CONCAT combined with slices and layers of convolution kernels to obtain a 320 × 320 × 32 image by reducing the image dimension to accelerate the training. SPP is a pooling layer that keeps the image input from being limited by the input size while maintaining integrity. Neck consists of CBL, Upsample, CSP2_X, and other models; it is a series of feature aggregation layers that combines image features and generates feature pyramid (FPN) and path aggregation (PAN) networks. YOLOv5 retains the CSP1_X structure of the YOLOv4 version CSPDarknet-53 and then adds the CSP2_X structure to reduce the model size and extract increasingly complete image features. Head uses GIoU Loss as the loss function for bounding boxes.

The loss function in YOLOv5 is divided into three types: bounding box regression, confidence, and classification losses. The loss function is used to describe the predicted and true values of the model to indicate the degree of difference. The loss function of the YOLOv5 model is expressed as follows:(1)$$L_{GIOU}=\sum_{i=0}^{S^{2}}\sum_{j=0}^{B}I_{ij}^{obj}\left[1-IOU+\frac{A^{c}-U}{A^{c}}\right]$$
where $S^{2}$ represents the number of grids and *B* is the number of bounding boxes in each grid. The value of $I_{ij}^{obj}\;$ is equal to 1 when an object exists in the bounding box; otherwise, it will be 0.
(2)$$L_{conf}=-\overset{S^{2}}{\underset{i=0}{\sum }}\overset{B}{\underset{j=0}{\sum }}I_{ij}^{obj}\left[{\overset{⌢j}{C}}_{i}log\left(C_{i}^{j}\right)+\left(1-{\overset{⌢j}{C}}_{i}\right)log\left(1-C_{i}^{j}\right)\right]\;-\lambda_{noobj}\sum_{i=0}^{S^{2}}\sum_{j=0}^{B}I_{ij}^{noobj}\left[{\overset{⌢j}{C}}_{i}log\left(C_{i}^{j}\right)+\left(1-C_{i}^{j}\right)log\left(1-C_{i}^{j}\right)\right]$$
where ${\overset{⌢j}{C}}_{i}$ is the predicted confidence of the bounding box of *j* in the grid of *i*, $C_{i}^{j}$ is the true confidence of the bounding box of *j* in the grid of *i*, and $\lambda_{noobj}\;$ is the confidence weight when objects are absent in the bounding box.
(3)$$L_{class}=-\sum_{i=0}^{S^{2}}I_{ij}^{noobj}$$
(4)$$\sum_{c\in \mathrm{classes}}\left[{\overset{⌢j}{P}}_{i}\left(c\right)log\left(P_{i}^{j}\left(c\right)\right)+\left(1-{\overset{⌢j}{P}}_{i}\left(c\right)\right)log\left(1-P_{i}^{j}\left(c\right)\right)\right]$$
where ${\overset{⌢j}{P}}_{i}\left(c\right)$ is the probability that the detected object is predicted to be the category and $P_{i}^{j}\left(c\right)$ is the probability that it actually belongs to the category.

## 3. Results

The batch size was set to 16, and value curves of the training and validation sets were determined after 300 iterations of training the loss function. Detailed information on the original image is presented in Appendix A). The confusion matrix of the developed models in this study is presented in Table 1. Among the 384 test images, 301 images were correctly predicted in the RGB model while 274 images were correctly identified in the HSI model. The results were analyzed using several indicators, including precision, recall, specificity, accuracy, and F1 score, to evaluate the detection performance of the YOLOv5 network (see Appendix A for the detailed information on different equations of indicators). The obtained results are listed in Table 2 (see Appendix A for the heatmap of the confusion matrix). The respective accuracies of the RGB and HSI models of 0.792 and 0.787 indicated their similarity. The more evident SCC category in the HSI model presents a more significant improvement with other labels than the original image category SCC. This finding demonstrated the high degree of completeness of the model in the learning of lesion features. It can be seen that the accuracy rate of both the RGB model and the HSI model are similar because the number of images used are comparatively lesser. However, it can be proved from this study that the HSI-based conversion algorithms which have the capability to convert the RGB images to HSI images have the potential to classify and detect skin cancers. Although most of the results are similar, this research has significantly improved the recall rate and specificity of the SCC category. This study has also improved the specificity also known as the true negative rate for the SK category.

The future scope of this study is to implement the hyperspectral imaging conversion technique to capsule endoscopy. In a capsule endoscopy, there is no space to add an additional narrow band imaging filter which improves the performance of detecting cancers. The same algorithm can be used to convert the RGB images to HSI images. Another advantage of this algorithm is that it reduces the cost of the hyperspectral dermatologic acquisition system, which is not cost-effective. Therefore, this method can be implemented on the RGB images taken to detect skin cancers. In addition, video dermatoscopes combined with AI software can also be used for automatic diagnosis.

## 4. Conclusions

Skin cancer images are primarily classified into three categories of SCC, BCC, and SK using YOLOv5 based on the CNN architecture. Lesion categories, including RGB and hyperspectral datasets, are utilized to create a confusion matrix and calculate the precision, recall, specificity, precision, and F1-score values, which are used as classification indicators. The experimental results showed that the SCC category in the HSI model can successfully learn characteristic features. The HSI classification model presents a better detection effect than the original RGB image. The recall rate improved from 0.722 to 0.794. The HSI model can extract detailed spectral bands, reduce the noise from the image, and highlight lesion features. Because some symptoms of BCC are similar to SK, it is easy to be confused, so the recall rate is the lowest. Compared with previous versions, YOLOv5 is simpler to use because the input image can be automatically scaled to the required size. The relatively small model structure achieves an accuracy similar to that of the previous version at a faster speed. The gain can be used in the subsequent designs of the experiment to improve the accuracy in future investigations.

## Figures and Tables

**Figure 1 jcm-12-01134-f001:**
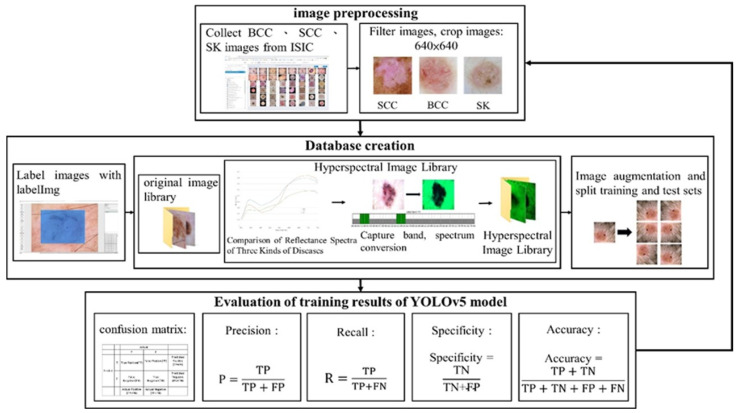
Overall experimental flow chart.

**Figure 2 jcm-12-01134-f002:**
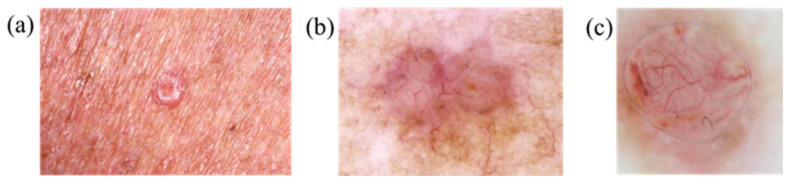
Basal cell carcinoma: (**a**) bright protrusions around the epidermis and (**b**,**c**) enlarged and frequently hemorrhagic lesions.

**Figure 3 jcm-12-01134-f003:**
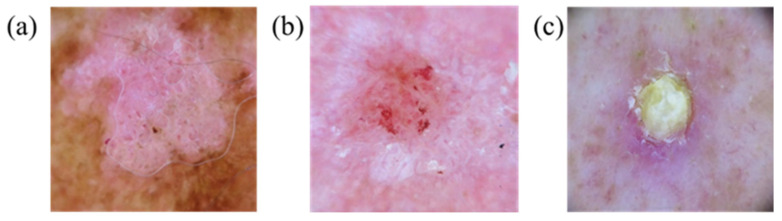
Squamous cell carcinoma appears as (**a**,**b**) red squamous plaques or (**c**) nodules.

**Figure 4 jcm-12-01134-f004:**
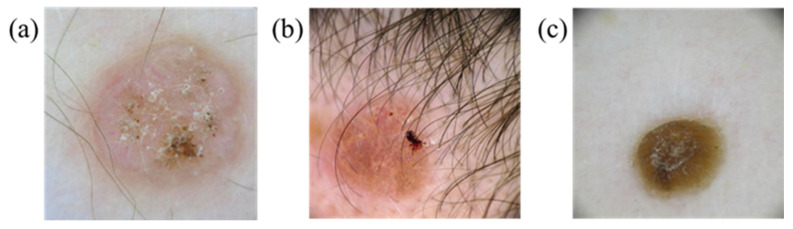
Seborrheic keratosis (**a**,**b**) due to hyperplasia of mutant epidermal keratinocytes and (**c**) black mass of keratinocytes.

**Figure 5 jcm-12-01134-f005:**
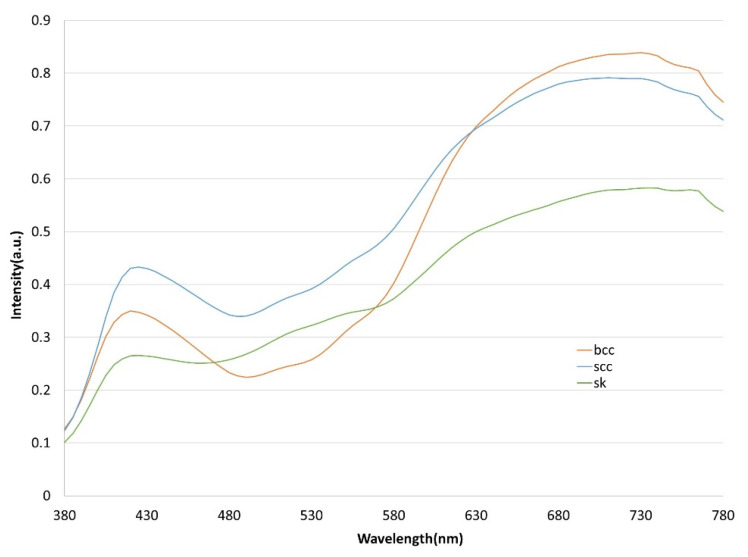
Normalized reflection spectra of BCC, SCC, and SK. The orange line represents BCC, the blue line represents SCC, and the green line represents SK.

**Figure 6 jcm-12-01134-f006:**
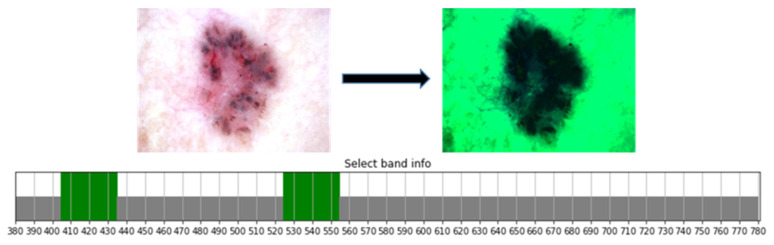
Bands 380–780, 405–435, and 525–555 nm between the 401 channels are selected and extracted.

**Table 1 jcm-12-01134-t001:** Confusion matrix for the YOLOv5 detection model.

Skin Disease		Results of the RGB Model			
		True			
		BCC	SCC	SK	Background FP
Predicted	BCC	133	7	8	45
	SCC	6	66	0	27
	SK	6	1	102	54
	Background FN	23	16	16	
**Skin Disease**		**Result of HSI Model**			
		**True**			
		BCC	SCC	SK	Background FP
Predicted	BCC	102	4	19	74
	SCC	17	72	0	10
	SK	6	0	100	55
	Background FN	43	14	7	

**Table 2 jcm-12-01134-t002:** Performance results of YOLOv5 training.

RGB Model	Precision	Recall	Specificity	F1-Score	Accuracy
All	0.888	0.758	0.798	0.818	0.792
BCC	0.899	0.747	0.791	0.816	
SCC	0.812	0.722	0.833	0.764	
SK	0.954	0.805	0.71	0.873	
**HSI Model**	**Precision**	**Recall**	**Specificity**	**F1-score**	**Accuracy**
All	0.8	0.726	0.786	0.761	0.787
BCC	0.813	0.624	0.716	0.706	
SCC	0.746	0.794	0.878	0.769	
SK	0.841	0.76	0.764	0.798	

## Data Availability

The data presented in this study are available in this article.

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
