# Peer review of "Classification of Skin Cancer Using Novel Hyperspectral Imaging Engineering via YOLOv5"

_jcm, 2023, doi:10.3390/jcm12031134_

Round 1

Reviewer 1 Report

This paper present an idenification and classification skin cancer method based on hyperspectral imagning.

The dataset from ISIC library was used in this paper and it was divided into training and test sets.

YOLO v5 is used with only look once image applied to  train  the  model. 

In order to determine the  performance the authors was generated confusion matrix and five indicating parameters including precision, recall, specificity, accuracy, and  F1-score of  the  trained  model.

The authos present a recall  rate  improved  from  0.722  to 0.794.

The works is well presented, and the database used is a brenchmark used in related work.

YOLO is a system used showing good improvements in this work.

Reviewer 2 Report

The manuscript presents a method for rapid detection and classification of skin cancer based on a combination of hyperspectral imaging and neural networks.  It is worth noting that the current title of the article may be misleading, as the authors only analyse images of different types of the skin cancer already identified.
The main concern regarding this article is why has the proposed approach not been tested on images of other skin lesions and benign skin formations? In the current version, this work only shows the potential of the method for classifying different types of skin cancer, but by no means for identifying it.

It is also worth noting that the introduction is very sparse on the possibilities of hyperspectral imaging in the diagnosis of skin cancer, even though there are numerous papers on this topic.

There are errors and typos in the text of the paper (e.g. the basal cell carcinoma on L40 is misspelled).

Reviewer 3 Report

In this paper, the authors adopt hyperspectral imaging (HSI) and deep learning for identification and classification of skin cancer. However, there are some problems that should be pointed out.

1. The title of this paper can be changed to "Identification and Classification of Skin Cancer Using Novel Hyperspectral Imaging Engineering via Deep Learning" or "Identification and Classification of Skin Cancer Using Novel Hyperspectral Imaging Engineering via YOLOv5".

2. The performance of YOLOv5 on this task should be compared with other deep learning architectures on computer vision, such as ResNet or DenseNet. Otherwise, the results cannot demonstrate the superiority of YOLOv5.

3. In Table 1, the performance of RGB model is better than HSI model, so the results cannot demonstrate the superiority of HSI.

4. The confusion matrix should be presented as heatmap rather than table.

5. The authors should analyze the results more adequately, especially paying attention to the interpretability of the results on medical view, such as adding case studies.

Reviewer 4 Report

Very interesting paper reporting a novel technology.

I only suggest to improve some parts to make the message clearer to the physicians readers. especially in the discussion part:

1- describe the benefits taht this technology will add to the existing diagnostic support tools (videodermatoscopes with AI software for automatic diagnosis..) in terms of diaghnostic accuracy and/or prognostic prevision.

2- it is not clear this statement:

"BCC exhibits the minimum recall rate because its 200 characteristic features are similar to those of SK." From a dermoscopic point of view, this is not possible. So please clarify that is due to non-dermoscopic features, (e.g., from a biological point of view/chemical composition...)

3.- typos

Line 81: images and a training validation set of 168, 90, and 126

 Line 121: Wavelengths in the range of  405-435 and 525–555 nm are used

 Line 137: The function of [] is to slice the input image

Round 2

Reviewer 3 Report

In general, the authors have corrected their paper following the reviewers' suggestions. The only thing that I need to point out, is that Table 2 should be put into one page. I also suggest the authors to combine Table 2 and Table S1, i.e. annotate the values on the heatmap.